# Multi-Omics Analysis Decodes Biosynthesis of Specialized Metabolites Constituting the Therapeutic Terrains of *Magnolia obovata*

**DOI:** 10.3390/ijms26031068

**Published:** 2025-01-26

**Authors:** Megha Rai, Amit Rai, Towa Yokosaka, Tetsuya Mori, Ryo Nakabayashi, Michimi Nakamura, Hideyuki Suzuki, Kazuki Saito, Mami Yamazaki

**Affiliations:** 1Graduate School of Pharmaceutical Sciences, Chiba University, Chiba 260-8675, Japan; megha@illinois.edu (M.R.); ju.zero19990407@gmail.com (T.Y.); michimi@chiba-u.jp (M.N.); 2Crop Sciences, University of Illinois Urbana, Champaign, IL 61801, USA; amitrai9@illinois.edu; 3RIKEN Center for Sustainable Resource Science, Yokohama 230-0045, Japan; tetsuya.mori@riken.jp (T.M.); roy.nakbayaski@gmail.com (R.N.); kazuki.saito@riken.jp (K.S.); 4Kazusa DNA Research Institute, Chiba 292-0818, Japan; h_suzuki@hirata.co.jp; 5Plant Molecular Science Center, Chiba University, Chiba 260-8675, Japan

**Keywords:** neolignans, integrative omics, metabolomics, natural products, laccases

## Abstract

*Magnolia obovata* is renowned for its unique bioactive constituents with medicinal properties traditionally used to treat digestive disorders, anxiety, and respiratory conditions. This study aimed to establish a comprehensive omics resource through untargeted metabolome and transcriptome profiling to explore biosynthesis of pharmacologically active compounds of *M. obovata* using seven tissues: young leaf, mature leaf, stem, bark, central cylinder, floral bud, and pistil. Untargeted metabolomic analysis identified 6733 mass features across seven tissues and captured chemo-diversity and its tissue-specificity in *M. obovata*. Through a combination of cheminformatics and manual screening approach, we confirmed the identities of 105 metabolites, including neolignans, such as honokiol and magnolol, which were found to be spatially accumulated in the bark tissue. RNA sequencing generated a comprehensive transcriptome resource, and expression analysis revealed significant tissue-specific expression patterns. Omics dataset integration identified T12 transcript module from WGCNA being correlated with the biosynthesis of magnolol and honokiol in *M. obovata*. Notably, phylogenetic analysis using transcripts from T12 module identified two laccase (Mo_LAC1 and Mo_LAC2) and three dirigent proteins from the DIR-b/d subfamily as potential candidate genes involved in neolignan biosynthesis. This research established omics resources of *M. obovata* and laid the groundwork for future studies aimed at optimizing and further understanding the biosynthesis of metabolites of therapeutic potential.

## 1. Introduction

*Magnolia obovata,* commonly known as “Hōnoki” in Japanese, belongs to the Magnoliaceae family and is native to eastern Asia, including Japan, China, and Korea [1]. It is a deciduous tree grown widely in Europe and America for ornamental purposes and has been used as traditional medicine in Asia for centuries [2]. In addition to its medicinal uses, plants extract from genus *Magnolia* are also used as dietary supplements and cosmetic products [3,4]. From nearly 220 known *Magnolia* species, *M. obovata* and *M. officinalis* are the most commonly used species used in traditional Chinese and Japanese herbal medicine and are incorporated in the Japanese pharmacopeia [4]. The bark of *M. officinalis* and *M. obovata*, also known as Houpu and Honoki, respectively, has been used in several medicinal formulations, including Banxia Houpu Tang and Hange-Kouboku-to, among others [4,5]. Two of the prescriptions in Japan, Sai-Boku-to and Hange-Kouboku-to, contain *Magnolia obovata* bark and are still used in modern clinical practice [4]. The bark tissue of *M. obovata* is highly regarded for its anti-cancer, anti-inflammatory, antioxidant, anti-stress, anti-anxiety, anti-depressant, anti-Alzheimer, and anti-stroke effects [3,6]. Additionally, it has also been shown to have vascular relaxation, hepatoprotective, and anti-diarrhea effects [4]. While the bark has been traditionally used for medicinal purposes, several studies have also reported the pharmacological activities of the floral buds and leaves of *Magnolia* plants, providing a basis for investigations into the use of other plant tissues for its use as drugs [4,7].

The medicinal properties of *M. obovata* are attributed to the presence of a diverse array of bioactive compounds, such as lignans, neolignans, terpenoids, alkaloids, and flavonoids [2,8,9]. To date, around 255 biologically active substances have been isolated from different tissues of the genus *Magnolia*, of which neolignans, namely magnolol and honokiol, are considered the principal active ingredients [4,10]. It is believed that magnolol and honokiol are the key metabolite constituents responsible for the therapeutic effects of *Magnolia* plants [2]. However, only 31 metabolites have been reported specifically for *M. obovata* in the KNApSAck database [11]. Despite the medicinal importance of *M. obovata*, the biosynthetic pathways and regulatory mechanisms underlying the biosynthesis of therapeutic compounds in this plant species remain largely unexplored. Elucidating the biosynthetic genes and pathways associated with bioactive compounds in *M. obovata* can provide valuable insights into the molecular basis of its medicinal properties and enable potential applications in medicinal and pharmaceutical industries. The utilization of multi-omics resources has been demonstrated as an effective means to identify natural variants with desired properties, constituting an essential element for the sustainable production of bioactive metabolites from medicinal plants.

The surge in advanced omics technologies, driven by reduced cost for data generation, enhanced computational power for data integration and analysis, and high-throughput and high-resolution data generation capabilities. These advancements have emerged as powerful tools for studying the biosynthesis of bioactive compounds in plants [12]. In the absence of genomic resources, RNA-seq based de novo transcriptome assembly has been established as a valuable resource for understanding the biosynthesis of important specialized metabolites in non-model plant species such as medicinal plants [13,14,15,16]. Untargeted metabolome analysis insights into the accumulation patterns of metabolites including bioactive compounds in different plant tissues, while transcriptome analysis provides information on the expression patterns of molecular elements involved in the biosynthesis of these metabolites [12,13]. While the individual analysis of omics datasets can provide systems characteristics at the metabolite and gene expression levels, integration of multi-omics datasets assists in identifying metabolite-gene associations and to prioritize key enzymes by removing the false positives for functional characterization, thereby providing a comprehensive understanding of their biosynthetic pathway in plants [17,18,19,20].

In this study, we aimed to unravel the biosynthetic genes associated with key metabolites of therapeutic importance by adopting an integrated metabolome and transcriptome analysis approach by using seven tissues of *M. obovata*, including young leaf, mature leaf, stem, bark, central cylinder, floral bud, and pistil. Using untargeted metabolomics, we investigated the metabo-constituents of different tissues of *M. obovata*. Transcriptome analysis identified the ongoing biological processes and the actively expressed transcripts. Furthermore, module-based analysis and integration of the -omics datasets facilitated the identification of potential transcripts involved in the biosynthesis of neolignans, namely magnolol and honokiol. Our findings provide valuable insights into the molecular basis of the biosynthesis of therapeutic compounds in *M. obovata*. This study presents comprehensive -omics resources for *M. obovata* and will help identify candidate therapeutic metabolites in different tissues for future validation, structural determination, and functional characterization of associated biosynthetic enzymes.

## 2. Results

### 2.1. Metabo-Constituents of Magnolia obovata and Its Chemo-Diversity Revealed by Untargeted Metabolomics

Several studies have reported a plethora of metabolites being present in *Magnolia* plants [3,4,8]. However, most of these studies have focused on investigating the components present in the bark of *Magnolia* plants and, to some extent, in the fruits, leaves, and flowers [21,22,23,24,25,26,27,28,29]. To establish an accurate metabolome representation for *M. obovata* covering multiple tissues, we performed untargeted metabolome profiling using seven tissues, namely young leaf, mature leaf, stem, bark, central cylinder, floral bud, and pistil (Figure 1). The acquired raw metabolome datasets were pre-processed, aligned, and metabolite features were extracted using MS-DIAL [30], resulting in 6733 mass features in the positive ionization mode (Appendix A).

Principal component analysis (PCA) using the normalized intensities of mass features showed metabolome datasets of seven tissues of *M. obovata* into five distinct groups (Figure 2a). Bark, the medicinally important tissue, formed a distinct group and was separated along the PC1 axis from the rest of the tissues. The young leaf, mature leaf, and stem were grouped together and separated from the floral bud and pistil along the PC2 axis, which formed a distinct tissue-group by itself. The PCA showed tissue specific metabolome signature being captured by the untargeted metabolome profiling of seven tissues of *M. obovata*, which is an ideal feature for a high-quality metabolome resource effectively capturing the tissue-specific chemo-diversity. Furthermore, we mapped metabolite features against the KEGG [31] and KNApSAck [11] databases to assign chemical identities to the acquired mass features. In total, 1705 and 553 mass features were assigned with chemical identities to a corresponding KNApSAck and KEGG ID, respectively (Appendix A). Several of the secondary metabolite biosynthesis pathways, including “Isoquinoline alkaloid biosynthesis”, “Biosynthesis of various plant secondary metabolites”, “Phenylpropanoid biosynthesis”, and “Flavonoid biosynthesis”, were among the top 15 KEGG pathways being represented based on the number of metabolites being mapped (Figure 2b). These results suggest that *Magnolia* tissues have a much bigger diversity of metabolites compared to what is known to date as revealed by the untargeted metabolomics in this study. Therefore, our study opens new avenues for the structural validation and identification of candidate metabolites as well as exploring tissue-wise metabolite accumulation for specialized metabolites to hypothesize phytochemical’s role in plant physiology and further providing a basis for investigations into the use of tissues other than the bark of *M. obovata* as a potent drug source.

### 2.2. Distinctive Metabolic Signatures of Magnolia obovata Tissues

As a proof of concept to use the established metabolome resource of *M. obovata* for metabolite discovery and analysis across different tissues, we used a combination of computational chemo-informatics approach, primarily driven through MS-FINDER [32], together with manual inspection and validation using MS and MS/MS-based fragmentation data to confirm the identity of 105 metabolites across seven tissues of *M. obovata* (Appendix A). The accumulation of these 105 metabolites showed a tissue-specific trend. To explore hidden signature of metabolites accumulation trend, we adopted a Weighted correlation network analysis (WCNA)-based approach using accumulation levels of 105 metabolites. WCNA is a weighted correlation-based method to define a network from high-dimensional -omics data by grouping the highly correlated variables into a signature module [33]. WCNA classified 97 of the 105 metabolites into five distinct modules (Figure 3), while eight remained unassigned. We identified 13, 13, 23, 38, and 10 metabolites being assigned to the metabolite modules, namely M1, M2, M3, M4, and M5, respectively (Appendix A). Module M1 was highly accumulated in the young leaf and mature leaf of *M. obovata*, while metabolite module M2 was accumulated majorly in the mature leaf. The metabolite modules M1 and M2 were dominated by the presence of flavonoids and flavonoid glycosides, including quercetin, rutin, quercitrin, kaempferol, quercetin-4′-O-glucoside, quercetin-3-O-rutinoside. Our results are concurrent with previous studies reporting the accumulation of flavonoid-type metabolites leaves of *Magnolia* species [9]. The metabolite module M3 included several amino acids, including tryptophan, arginine, asparagine, proline, among others and was highly accumulated in the pistil tissues of *M. obovata*. The pistil, female reproductive organ, plays a crucial role in fertilization, and the accumulation of amino acids in the pistil may signify its role in facilitating successful fertilization and subsequent seed development in plants. Furthermore, magnaldehyde B, a known lignan of *Magnolia* species isolated mainly from bark tissue [9], was also highly accumulated in the pistil. Magnaldehyde B and its analogs have been reported to possess anti-tumor activity [9]. The metabolite module M4, representing the largest module with 38 metabolites, was highly accumulated in the bark tissue of *M. obovata*. Magnolol and honokiol, the principal bioactive compound of *Magnolia* species, were members of the M4 module, thus, corroborating with the tissue primarily used for medicinal purposes. The metabolite module M5, representing the smallest module, included several intermediates from the lignin biosynthetic pathway, including coniferyldehyde, 5-hydroxyconiferyl alcohol, and was highly accumulated in the stem of *M. obovata*. These lignin intermediates play crucial roles in the development and structure of plant stems. Metabolite module M5 also included loliolide, a monoterpene lactone used for treatment of central nervous system diseases. Our results clearly suggest that the WCNA of the MS^2^-validated metabolites successfully captured the unique metabolic profiles associated with each tissue type in *M. obovata*. Our metabolome analysis revealed that magnolol and honokiol, two principal bioactive compounds found in *Magnolia* species, exhibited highest accumulation in the bark of *M. obovata*. This underscores the significance of the bark of *M. obovata* for medicinal purposes. Nonetheless, the presence of significant flavonoids such as rutin and quercitrin, known for their antioxidant and anti-tumor properties, along with lignans like magnaldehyde B and obovatal, and monoterpene lactones such as loliolide in tissues apart from the bark, implies the potential role of different tissues for medicinal purposes.

### 2.3. Genomic Features Characteristics Through De Novo Transcriptome Assembly of Magnolia obovata

The availability of comprehensive genomic datasets is fundamental for precise species selection, trait delineation, and the formulation of biotechnology-driven approaches to enhance desired characteristics. This includes the strategic identification of specific species variants exhibiting elevated levels of bioactive metabolites, considering factors such as genetic diversity and environmental influences. While high-quality genome assemblies of non-model medicinal plants are still scarce, the applications of high-quality RNA-seq-based de novo transcriptome assembly in identifying the potential biosynthetic genes have been well established [34,35,36,37,38]. Therefore, to establish a comprehensive transcriptome resource, capturing the diversity of active transcripts across multiple tissues, we performed RNA-sequencing for the same set of seven tissues of *M. obovata* as was used for establishing metabolome resource. The RNA samples from the seven tissues of *M. obovata* were used to synthesis individual cDNA libraries, and they were subsequently sequenced using Illumina HiSeq^TM^ 2000 platform, generating over 15.76 Gbp of total raw sequence data.

The raw data were preprocessed using Trimmomatic [39] to exclude adapters and low-quality reads in the de novo assembly process, which dropped less than 1% of the sequencing reads. It is widely known that different de novo transcriptome assembly algorithms, even though are based on similar working principles, result in different assembled transcripts with varying lengths, transcriptome completeness and varying overlaps of transcriptome coverage. Merging individual RNA-seq assemblies from different assemblers leverages the strengths of each tool, resulting in a more comprehensive and accurate transcriptome representation. Previous studies have shown an advantage of using a hybrid de novo transcriptome assembly approach to capture diversity of transcripts with near to complete full-length transcripts for further functional characterizations [40,41]. Therefore, in order to achieve a high-quality transcriptome assembly for *M. obovata*, we adopted a hybrid assembly approach using the individual transcriptome assembly derived from three popular assemblers, namely CLC genomics workbench (https://www.qiagenbioinformatics.com/, accessed on 8 May 2022), Trinity v3.0 [42], and SOAPdenovo-Trans [43]. The resulting three de novo transcriptome assemblies were merged, followed by removing the redundancy using CD-HIT-EST [44] (Table 1). The hybrid assembly for *M. obovata*, thus obtained, included 284,165 transcripts with an average length and N50 value of 762 bp and 1280 bp, respectively. The length distribution of the assembled transcripts varied between 196 and 16,986 bp (Appendix A). To evaluate the completeness of our hybrid RNA-seq assembly, we employed BUSCO (Benchmarking Universal Single-Copy Orthologs), which assesses assembly completeness by mapping transcript sequences against a set of conserved single-copy orthologs [45]. The results demonstrated a high level of completeness in our hybrid RNA-seq assembly, with 98.82% of the BUSCO groups having complete gene representation (single-copy or duplicated), while 1.18% are fragmented, and 0% are missing (Appendix A). Consistent with previous studies, our results showed improved assembly stats for the hybrid de novo transcriptome assembly approach compared to the individual de novo transcriptome assemblies [14,18,19].

To characterize the de novo transcriptome assembly of *M. obovata*, we performed a Blastx-based homology search of the assembled transcripts against protein sequences deposited at the NCBI non-redundant (nr) database. Blastx-based searching resulted in the annotation of 105,980 transcripts, with 82,367 transcripts and 17,275 transcripts being associated with Gene Ontology (GO) annotation and GO mapping, respectively, (Appendix A). Notably, despite being phylogenetically distant, our analysis revealed significant sequence similarity with several plant species, with *Vitis vinifera*, *Populus trichocarpa*, and *Ricinus communis* emerging as the top three species based on the number of transcripts with high-sequence similarity with the assembled transcripts of *M. obovata* (Appendix A). This can be attributed to the conservation of core plant genes essential for fundamental biological processes, as well as genes involved in shared biosynthetic pathways. Evolutionary homology arising from a common ancestor may also play a role, with ancient gene families retaining high sequence similarity across angiosperms. This observation also highlights the limited representation of sequences from the genus *Magnolia* in existing NCBI databases, underscoring the significance of our study as a valuable resource of transcripts for future research and annotation efforts. The transcripts were assigned with annotations using the top hit from the Blastx search, providing valuable insights into the genetic underpinnings of its active biological processes including assignment of transcripts to different metabolic pathways, shedding light on potential biochemical pathways active within *M. obovata*. A total of 26,400 transcripts were assigned to 146 KEGG pathways, with the top 15 pathways based on the number of transcripts being assigned summarized in Appendix A. Among the top 15 pathways, starch and sucrose metabolism, glycolysis/gluconeogenesis, and drug metabolism-other enzymes were among the top three pathways, with 1158, 977, and 875 transcripts being mapped to these metabolic pathways, respectively. *M. obovata* is widely reported to contain phenylpropanoids and flavonoids as their major secondary metabolites [46,47]. KEGG pathway mapping assigned 478 and 306 transcripts to phenylpropanoid biosynthesis and flavonoid biosynthesis, respectively, with homologs of major known enzymes identified in our transcriptome datasets. Collectively, these findings establish a comprehensive framework for elucidating the biological significance of *M. obovata* transcripts and pave the way for future investigations into its molecular intricacies.

### 2.4. Transcripts Expression Analysis Reveals the Tissue-Specific Molecular Signatures in Magnolia obovata

To unravel the complex genetic makeup for a plant species, it is vital to understand as how the different genes expression and associations to different tissues orchestrates the plant’s growth, development, and responses to different biotic–abiotic challenges. Transcriptome expression analysis provides a comprehensive overview of the active biological processes, where the spatial expression and relative expression abundance of a given gene provides valuable hints for its potential functions. To calculate the qualitative expression level of each transcript across different tissues, the processed paired-end RNA-seq reads were mapped to the de novo transcriptome assembly of *M. obovata*, and normalized transcript expression in FPKM (Fragments Per Kilobase of transcript per Million mapped reads) was determined. Among the seven tissues of *M. obovata*, stem and bark exhibited the highest number of transcriptionally active transcripts, with 186,216 and 185,959 transcripts, respectively, displaying expression levels above FPKM > 0. On the other hand, floral bud and pistil displayed the lowest number of transcriptionally active transcripts, with 127,323 and 135,322 transcripts, respectively, (Appendix A). The differential distribution of transcriptionally active genes across tissues underscores the functional diversity and the requirement of specific sets of transcripts activity to achieve the specific functionality. A total of 69,697 transcripts were shared among all seven tissues, suggesting their potential role in the essential biological functions such as primary metabolic and core metabolic processes. Six of the seven tissues of *M. obovata* showed expression specific to the tissue for transcripts in the range of 2500–5250. Bark, on the other hand, showed specific expression for 24,242 transcripts (Appendix A), which co-incidentally is also the tissue that produces a range of specialized metabolites with bioactivity. The huge parity among the unique active transcripts between bark and the rest of the tissues suggest that bark includes a unique collection of transcripts for fine-tuning *M. obovata*’s response to environmental cues and to produces a diverse array of secondary metabolites.

To uncover the underlaying patterns and relationships within transcriptome datasets across seven tissues of *M. obovata*, we conducted unsupervised principal component analysis (PCA). The PCA results exhibited the presence of four distinct groups (Appendix A), with leaf tissues, including both young and mature leaves, clustered in quadrant III and segregated from other tissues along the PC1 axis. Along the PC2 axis, five tissues were partitioned into two quadrants, with bark and pistil grouping in quadrant I, exhibiting clear separation along the PC2 axis. Conversely, the floral bud, stem, and central cylinder were clustered in quadrant IV. Consistent with the PCA-based clustering, expression-based correlation analysis also identified all seven tissues being clustered into four clusters (Appendix A), wherein young leaf and mature leaf were clustered together, and bark and pistil tissues formed individual clusters. The floral bud, stem, and central cylinder were also clustered together, with the stem and central cylinder sharing a closer association. These analyses underscore the presence of tissue-specific transcriptomic signatures in *M. obovata*, highlighting their specialized functions. Further, the overlap of transcript expression across tissues seems to stem from their association at different stages of development.

### 2.5. Construction of Gene-Metabolite Co-Expression Network Uncovers Tissue-Specific Transcriptome Signature and Specialized Metabolites Biosynthesis Pathways in Magnolia obovata

The ability of weighted gene co-expression network analysis (WGCNA) to transform a comprehensive transcriptome expression dataset into a representation of interconnectedness, thus revealing the intricate relationships within, has been widely acknowledged, facilitating the exploration of interconnected biomolecules [18,48,49,50]. Rooted in the guilt-by-association concept, WGCNA operated on the premise that genes with correlated expression are often functionally related, participating in common pathways, regulatory mechanisms, or responses to environmental stimuli [33]. WGCNA helps in understanding the gene interactions and underlaying relationships within a biological system by grouping these co-expressed genes into modules, with genes within the same module tending to have similar biological functions. Leveraging this approach, it becomes feasible to predict novel genes involved in specialized metabolite biosynthesis pathways by exploring modules containing well-characterized ‘bait genes’ of interest. This approach has been proven effective in identifying functional genes associated with the biosynthesis of plants’ specialized metabolites in several model and medicinal plants [18,51,52,53,54]. Using a similar approach for *M. obovata*, we performed WGCNA using the highly expressed (FPKM > 5 in at least one of the seven tissues) and annotated transcripts and identified highly co-expressed transcript modules (Appendix A). WGCNA identified 12 transcript modules, of which T12 (5429 member transcripts) and T6(70 member transcripts) represented the largest and the smallest modules, respectively, (Appendix A). Notably, transcript expression patterns mirrored the observed accumulation profiles of metabolites, displaying tissue-specific trends. The transcript modules, namely T1, T2, T3, T8, T9, T11, and T12 showed tissue-specific expression trends for young leaf, pistil, floral bud, central cylinder, stem, mature leaf, and bark, respectively (Figure 4).

The integration of multi-omics datasets enables a broader exploration beyond gene expression patterns and metabolite accumulation by establishing gene-to-metabolite associations to reveal complex relationships among biomolecules. The integration allows for the examination of how metabolite accumulation patterns correlate with gene expression patterns within the same pathway. By combining gene co-expression data with metabolic profiling data, it becomes possible to identify genes whose expression patterns correlate with metabolite accumulation patterns in specific pathways, thereby suggesting their involvement in orchestrating observed metabolic shifts [12,13]. This integrated omics approach has facilitated the achievement of a refined and comprehensive prioritization of genes involved in a specific biosynthetic pathway of interest [12,18,19,28]. To implement the integrated omics strategy in *M. obovata,* we performed a Pearson-based correlation analysis using the average accumulation or expression value of the metabolites and transcripts assigned to the identified metabolome, and transcriptome modules, respectively. A notably high correlation (R^2^ > 0.9) was observed for four module pairs, M1–T11, M3–T2, M4–T12, and M5–T9 (Figure 5a, Appendix A). Among the metabolite–gene module pairs, M4–T12 was specific to the bark tissues and included magnolol and honokiol, the major constituents that give the characteristic medicinal properties to the *M. obovata*. Therefore, we further explored the transcript module T12. The top 15 KEGG pathways based on the number of transcripts from T12 module being mapped to the pathway database are summarized in Figure 5b. While the top KEGG pathways were mainly associated with primary metabolic pathways, transcripts within the T12 module also represented homologs of all genes involved in the biosynthesis of phenylpropanoids and flavonoids in *M. obovata* (Appendix A). Gene ontology enrichment analysis using annotation of transcripts from the T12 module showed enrichment of several metabolic processes associated with stress responses, including some response type processes, namely response to water deprivation and defense response to bacterium (Figure 5c).

### 2.6. Merging Omics Landscapes to Decode Neolignans Biosynthesis in Magnolia obovata

Neolignans, namely honokiol and magnolol, are the key bioactive constituents attributing to the characteristic medicinal properties of *M. obovata* [2]. Monolignols from the cinnamate/monolignol biosynthetic pathway are reported to be the intermediates in the biosynthesis of neolignans [55]. The biosynthesis of lignans is well characterized and is known to be initiated by oxidation of coniferyl alcohol to form corresponding radical through the enzymatic activity of oxidases such as laccase followed by the stereoselective dimerization catalyzed by dirigent proteins [56,57,58]. In contrast, our understanding of the biosynthesis pathway of neolignans is currently limited. However, several crude enzyme preparation-based studies have highlighted that similar to the lignans biosynthesis, polymerization of phenylpropanoid monomers also results in the formation of neolignans [55,59,60,61,62]. However, the oxidases and dirigent proteins involved in the biosynthesis of neolignans are still largely unknown.

Laccases (LACs), a class of multi-copper oxidases, are widely distributed across different kingdoms, including bacteria, fungi, plants, and animals [63,64]. Characterized by broad substrate specificity, laccases catalyze independently of any cofactors. Laccases mainly oxidize aromatic phenolics but also participate in the oxidizing reaction of diamines and benzenethiols through the reduction in molecular oxygen into water [63]. The radicals generated by laccase enzyme require the action of dirigent proteins (DIRs) that orient the two radicals correctly for them to undergo a regio- and stereo-selective coupling reaction [63]. These mechanisms of action have been majorly reported for the biosynthesis of lignans, ellagitannins, and stilbenoid dimers [63,65,66,67]. A recent study on Arabidopsis highlighted the role of laccase and dirigent proteins in the biosynthesis of seed-protective neolignans [68]. Considering the diverse substrate specificity observed in laccases, and reports of its novel functionalities observed in the biosynthesis of lignans, neolignans, and related compounds [63], we hypothesize that neolignans in *M. obovata* originate from the precursor coniferyl alcohol through a two-step process (Figure 6a). As the first step, laccases catalyze the generation of monolignol radicals. These radicals, stabilized through mesomeric forms, render multiple positions on the carbon skeleton reactive. Subsequently, in the second step facilitated by dirigent proteins, these radicals undergo dimerization, forming new C-C or C-O bonds.

The transcript module T12, which shared a high association with the metabolite module M4, which included the neolignans magnolol and honokiol, included homologs of all the enzymes involved in the biosynthesis of coniferyl alcohol. Therefore, LACs and DIRs included in the T12 module and sharing high co-expression with the other homologs of phenylpropanoid biosynthetic pathway are strong candidates with potential role in catalyzing the reaction to produce neolignans in *M. obovata.* Using homology-based annotation and functional characterization, we identified three LACs and seven DIRs, respectively, in the transcript module T12. To understand the type of reaction catalyzed by these LACs and DIRs in *M. obovata*, we performed a phylogenetic analysis of them with the functionally characterized laccases and dirigent proteins from other plant species.

Phylogenetic analysis of three of the laccases included in the T12 transcript module of *M. obovata* was performed with the 29 of the known laccases across different plant species, including *Arabidopsis thaliana, Populus trichocarpa, Zea mays,* among others (Appendix A). While one of the laccases of *M. obovata* formed a separate clade, two of the laccases clustered together with the laccase 5, laccase 12, laccase 3, and laccase 13 of *A. thaliana* and laccase 90 of *P. trichocarpa* (Figure 6b). While most of these Arabidopsis laccase remains uncharacterized, a recent study showed *AtLAC5* to oxidize coniferyl alcohol [68]. *PtLAC90* has also been shown to oxidize a wide range of substrates, including coniferyl alcohol [69]. Moreover, both laccase encoding transcripts in *M. obovata* contain the known functional multicopper oxidase domain. Therefore, inferring from our phylogenetic analysis, we propose these two laccases, namely *Mo_LAC1* and *Mo_LAC2*, as strong candidate genes potentially involved in the oxidizing of coniferyl alcohol, the first reaction towards the biosynthesis of neolignans in *M. obovata.*

To identify the potential DIRs involved in the second step of neolignans biosynthesis, we performed a phylogenetic analysis of seven of the DIRs of *M. obovata* with the known DIRs spanning across various plant species (Appendix A). Our comprehensive phylogenetic analysis provides invaluable insights into the relationships between various subfamilies of dirigent proteins with dirigent proteins of *M. obovata* from T12 module, therefore, shedding light on the complexity of neolignans biosynthesis. While one of the transcripts formed a separate clade (Figure 6c), notably, the six transcripts span two subfamilies, five to DIR-b/d subfamily and one with DIR-g subfamily, signifying multifaceted repertoire of dirigent proteins at play in *M. obovata*’s biochemical landscape. While less characterized, the DIR-g is a species-specific subfamily, and putatively participates in the biosynthesis of lignins [70,71], and the *M. obovata* transcript clustered together within the DIR-g subfamily may suggest its role in the biosynthesis of lignins. Most transcripts included in the T12 transcript module of *M. obovata* clustered within the DIR-b/d subfamily, making this subfamily a focal point of our investigation. The majority of the genes from the DIR-a subfamily of dirigent proteins is primarily associated with synthesizing pinoresinol lignans [71,72]. On the other hand, DIR-b/d proteins are more versatile and renowned for their remarkable substrate specificity, guiding the synthesis of a broader range of compounds [73]. It has also been reported that the DIR-b/d subfamily has expanded in many plant families, suggesting their role in the plant adaptive evolution under different biotic and abiotic stresses [71,73,74,75]. Therefore, five transcripts being grouped with the DIR-b/d subfamily may suggest their potential role in the biosynthesis of neolignans in *M. obovata*. Moreover, three of the five transcripts from the DIR-b/d subfamily had functional dirigent-like protein domain, thus, prioritizing their role in the biosynthesis of neolignans in *M. obovata*. Results from the phylogenetic analysis open doors to further exploration. Future research to delve deeper into the functional characterization of transcripts affiliated with different DIR subfamilies may help uncover how they influence neolignans biosynthesis in *M. obovata*.

## 3. Materials and Methods

### 3.1. Plant Materials

The *Magnolia obovata* plant was maintained at the medicinal plant gardens, Chiba University, under natural conditions (Figure 1). The seven tissues of *M. obovata*, including young leaf, mature leaf, stem, bark, central cylinder, floral bud, and pistil, were harvested and immediately frozen in liquid nitrogen. The tissues were stored at −80 °C until further use. For RNA extraction, tissues were homogenized to fine powder using mortar and pestle. For metabolites extraction, tissues were homogenized to fine powder using a mixer mill, MM300 (Retsch, Haan, Germany). Here, we adopted the split experimental design, where same tissues were split into two parts, each used for generating transcriptome and metabolome datasets.

### 3.2. Untargeted Metabolite Analysis for Seven Tissues of Magnolia obovata

Each of the homogenized tissues of *M. obovata* were freeze-dried and metabolites were extracted using 50 μL of 80% (*v*/*v*) LC–MS-grade methanol (Wako chemicals, Osaka, Japan) and 20% (*v*/*v*) LC–MS-grade water (Wako chemicals, Osaka, Japan) containing 2.5 μM of 10-camphoursulfonic acid (TCI, Tokyo, Japan), and 2.5 μM of lidocaine (TCI, Tokyo, Japan) per milligram of dry weight. After centrifugation using an Eppendorf 5417R centrifuge (Eppendorf SE, Hamburg, Germany) at 17,800× *g* for 10 min, the supernatant was subsequently processed using Oasis HLB μElution plate (Waters Inc., Milford, MA, USA) for removal of lipids and impurities, and the MS and MS^2^ datasets were acquired in the positive and negative ionization mode as described previously [18]. Peaks were aligned using MS-DIAL v4.80 with its default parameters. Peak normalization was conducted by comparison with an internal standard (lidocaine). Metabolites were annotated using MS-FINDER v3.52 and using *m*/*z* values of diterpene alkaloids being reported previously with tolerance 10 ppm. Putative chemical identities were assigned to the acquired metabolite peaks by mapping onto the KNApSAcK database and Kyoto Encyclopedia of Genes and Genomes (KEGG) pathway database. The MS^2^-based validation of metabolites was conducted as reported previously [18].

### 3.3. RNA Extraction, and cDNA Library Preparation

Frozen tissues of *M. obovata*, the same tissues used for untargeted metabolome analysis, were used for RNA extraction and cDNA library preparation. RNA extraction, RNA integrity analysis, mRNA sample preparation, fragmentation of isolated mRNA, and cDNA library preparation for Illumina sequencing were performed as described previously [16].

### 3.4. Illumina Sequencing

The cDNA library thus prepared for each tissue of *M. obovata* was sequenced using an Illumina HiSeq™ 2000 sequencer (Illumina, Inc., San Diego, CA, USA), and paired-end reads were obtained with an average length of 101 bps. Preparation and shearing of mRNA, cDNA library preparation, and sequencing were performed at Kazusa DNA Research Institute, Chiba, Japan. The raw sequence reads for all seven tissues of *M. obovata*, their expression value and the de novo transcriptome assembly used in this study have been deposited in NCBI’s Gene Expression Omnibus (GEO) and are available at GEO series accession number GSE275868.

### 3.5. RNA-seq Raw Reads Pre-Processing, De Novo Transcriptome Assembly, and Functional Classification of Transcripts

Raw sequencing reads thus generated were preprocessed through the Trimmomatic program v0.39 [39] to remove adaptor sequences, short reads, reads with ambiguous ‘N’ base > 5%, and low-quality reads (Phred score < 30). Paired-end processed reads, as well as unpaired high-quality reads that lost their corresponding sequence partner due to Trimmomatic filtering, for all seven tissue types were combined to build a de novo transcriptome assembly of *M. obovata*. The de novo transcriptome assembly resulting from the Trinity program v3.0 [42] was further processed through the CD-HIT-EST program v4.8.1 [44] for sequence redundancy removal, and it was and subsequently used for annotation and characterization. The transcripts obtained were, thus, used for the read alignment and abundance estimation in individual tissues of *M. obovata* using Bowtie 2.0 v2.3.5.1 [76] and RSEM v1.3 [77], respectively. Transcript expression was calculated in terms of Fragments per Kilobase exon per million mapped fragments (FPKM). Correlation analysis was performed for all seven tissues of *M. obovata* through the DeSeq2 program v3.17 [78] of the R-package. The completeness and quality of the *M. obovata* de novo transcriptome assembly were assessed using Benchmarking Universal Single-Copy Orthologs (BUSCO) [45] analysis against the viridiplantae_odb10 database, conducted through OmicsBox v3.1.2 (https://www.biobam.com/omicsbox; access date 25 January 2025).

The annotation of de novo transcriptome assembly of *M. obovata* was performed using a Blastx-based homology search against the NCBI-nr database with an E-value cutoff < 10^−5^, and the top Blastx hit was used to assign sequence description and putative functionality to the transcripts. Furthermore, OmicsBox v3.1.2 was used to obtain GO terms, EC number, and KEGG pathway-based annotation for the transcripts of *M. obovata*. For the identification of transcripts involved in the biosynthesis of diterpene alkaloid, the annotated transcriptome assembly was screened for the homologs of the associated enzymes, and candidate transcripts were selected that had a length greater than 500 bps and sequence similarity with top Blastx hit of over 70%.

### 3.6. Module Construction Using Omics Dataset of Magnolia obovata

For metabolome data, 105 MS^2^-validated metabolites were used for the module construction. For the identification of modules in the transcriptome datasets, we first filtered the assembled transcripts using the following parameters: (i) length > 500 bps and similarity > 70%, and (ii) FPKM > 5 in at least one of the seven tissues, resulting in narrowing down of 25,728 transcripts, which were subsequently used for WGCNA in R. Hierarchical clustering based on the topological overlap (TO) similarity was used to obtain network modules, and the highly similar modules (dissimilarity < 0.25) were merged to prevent over-splitting of the modules. Module eigenvalues, representing the overall profile of the module, were calculated for each of the identified modules.

### 3.7. Integrative Omics Analysis Through Correlation of Metabolite and Transcript Modules of Magnolia obovata

The average metabolite levels and the FPKM values of transcripts clustered in the individual metabolite and transcript modules, respectively, were used for correlation analysis. A Pearson correlation analysis was performed to calculate correlation coefficient between each metabolite and transcript module and generate a pairwise Pearson correlation matrix. The correlation relationships between metabolites and transcripts were visualized using Hatmap2.0 in the R package.

### 3.8. Phylogenetic Analysis of Candidate Laccase and Dirigent Proteins in Magnolia obovata

The transcripts of *M. obovata* annotated as laccase and dirigent proteins, included in transcript module T12 and having a sequence length of over 500 bps as well as having sequence similarity with the top Blastx hit of 70% were selected for phylogenetic analysis. These selected transcripts were translated to their corresponding protein sequences by selecting the translation frame that resulted in the longest amino acid sequence while starting with methionine using OmicsBox v3.1.2. Protein sequences of candidate transcripts and previously known laccase and dirigent proteins involved in the biosynthesis of specialized metabolites in other plant species were aligned using the MUSCLE program v3.8.31 [79], and evolutionary distances were inferred using the Neighbor-Joining method [80] with bootstrap values obtained after applying 1000 replications using the MEGA X program v10.2.6 [81].

## 4. Conclusions

This study significantly enhances our understanding on the biosynthetic pathways of neolignans in *Magnolia obovata*, underscoring the complex interplay between metabolite accumulation and gene expression across various tissues. In this study, we established high-quality metabolome and transcriptome resources for *M. obovata*, which holds a unique position within the diverse *Magnolia* genus for its extensive pharmacological relevance in various traditional medicinal practices. The multi-omics approach focused on metabolomics and transcriptomics integration has provided valuable insights into the genetic and molecular underpinnings of this medicinal plant. By focusing on seven tissues, namely young leaf, mature leaf, stem, bark, central cylinder, floral bud, and pistil, we aimed to gain insight into the biosynthesis of specialized metabolites with therapeutic properties in *M. obovata*. The establishment of a robust omics resource revealed distinct tissue-specific accumulation patterns of metabolites, particularly in bark tissue, highlighting the adaptive strategies employed by this species. Through comprehensive untargeted metabolome and transcriptome profiling, we identified candidate genes associated with the biosynthesis of key bioactive compounds pivotal to the medicinal properties of *M. obovata*, including honokiol and magnolol. The integration of gene expression data with metabolite profiles facilitated the identification of candidate genes associated with neolignan biosynthesis, particularly enzymes that code for laccases and dirigent proteins. Phylogenetic analyses of these enzymes provided insights into their evolutionary relationships and potential roles in catalyzing specific biosynthetic reactions. The identification of Mo_LAC1 and Mo_LAC2 as strong candidates for oxidizing coniferyl alcohol lays a foundation for future functional characterization studies. Overall, this research not only clarifies the molecular mechanisms underlying neolignan biosynthesis *in M. obovata* but also offers valuable resources for the exploration of secondary metabolism in medicinal plants. The findings pave the way for further investigations into the regulatory networks governing these pathways, contributing to the broader understanding of plant biology and the potential development of natural products with therapeutic applications. Our findings also lay the groundwork for future endeavors in harnessing the therapeutic potential of *M. obovata*.

## Figures and Tables

**Figure 1 ijms-26-01068-f001:**
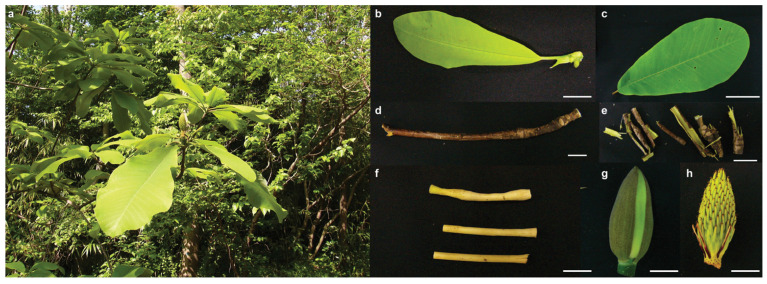
Tissues used for the multi-omics study on the biosynthesis of specialized metabolites in *Magnolia obovata*. (**a**) Plant growing under natural conditions in the herb garden, Chiba University; (**b**) Young leaf; (**c**) Mature leaf; (**d**) Stem; (**e**) Bark; (**f**) Central Cylinder; (**g**) Floral Bud; and (**h**) Pistil; the bars represent 1 cm.

**Figure 2 ijms-26-01068-f002:**
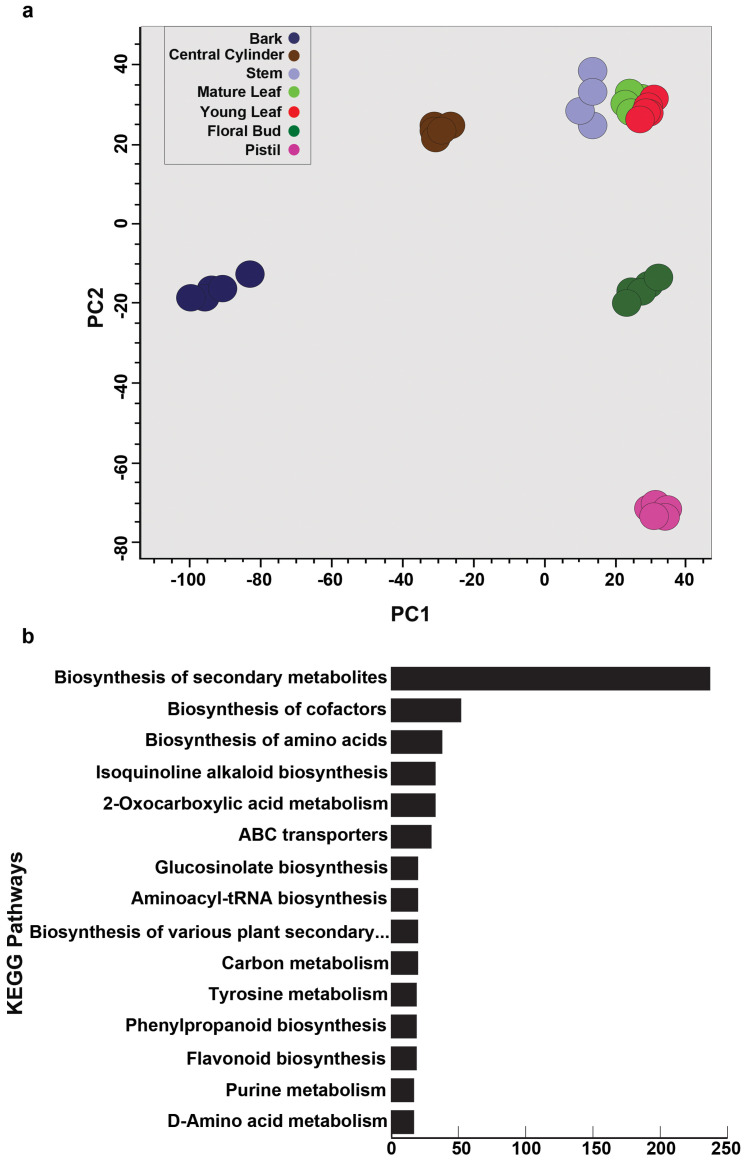
Untargeted metabolome analysis for seven tissues of *Magnolia obovata*. (**a**) Unsupervised Principal Component Analysis (PCA) using the metabolome datasets acquired for the seven tissues of *M. obovata*. (**b**) Top 15 KEGG pathways based on the number of assigned metabolites. The mass-features were mapped to the KEGG database, and the chemical identities were assigned based on the m/z similarity with a mass-error window of ±10 ppm. Furthermore, the KEGG pathways related to the assigned metabolites were extracted, the top 10 of which are shown here.

**Figure 3 ijms-26-01068-f003:**
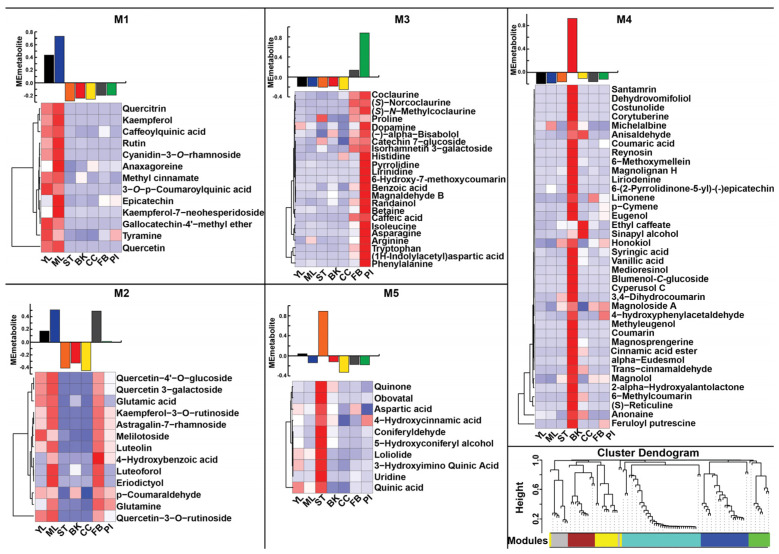
WCNA and accumulation pattern of MS^2^ confirmed metabolites in *Magnolia obovata*. The bar plots show the module eigenmetabolite of MS^2^-confirmed metabolites. The relative accumulation levels of metabolites included in each module are also shown by the heat maps above each bar plot. The dendrogram represents MS^2^-confirmed metabolites used for WCNA and clustering of module eigenmetabolites. Abbreviations: YL: Young leaf, ML: Mature Leaf, ST: Stem, BK: Bark, CC: Central cylinder, FB: Floral Bud, and PI: Pistil.

**Figure 4 ijms-26-01068-f004:**
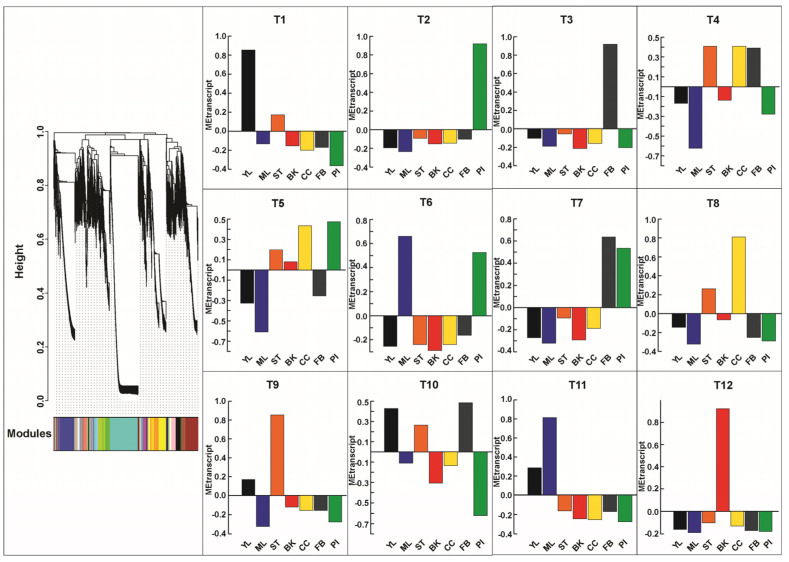
WCNA of the assembled transcripts in *Magnolia obovata*. The bar plots show the module eigentranscript of assembled transcripts. The relative accumulation levels of metabolites included in each module are also shown by the heat maps above each bar plot. The dendrogram represents expression level of transcripts used for WCNA and clustering of module eigentranscripts. Abbreviations: YL: Young leaf, ML: Mature Leaf, ST: Stem, BK: Bark, CC: Central cylinder, FB: Floral Bud, and PI: Pistil.

**Figure 5 ijms-26-01068-f005:**
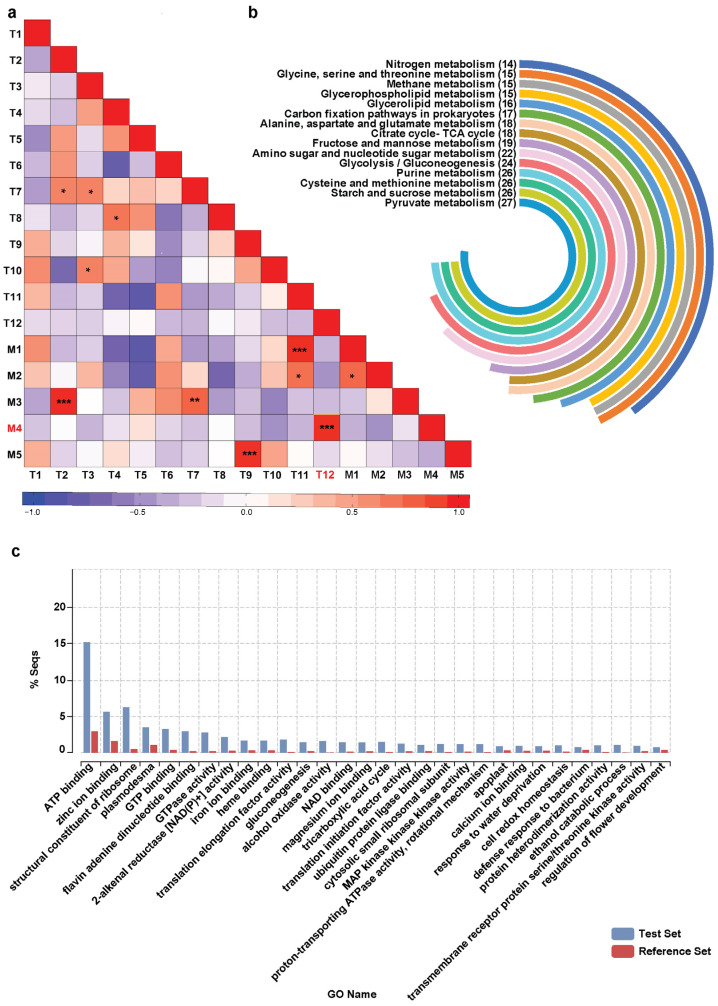
Module based integration of the metabolome and transcriptome datasets. (**a**) Pearson correlation coefficient between each MetM and TransM module pair. The red font highlights the transcript-metabolite module pair of interest. *** pairs with PCC > 0.9, ** pairs with PCC > 0.8, * PCC > 0.7. (**b**) The top 15 KEGG pathways based represented by transcripts included in T2 module based on the number of transcripts being mapped. (**c**) Gene Ontology (GO) enrichment analysis of the transcripts included in the T12 module.

**Figure 6 ijms-26-01068-f006:**
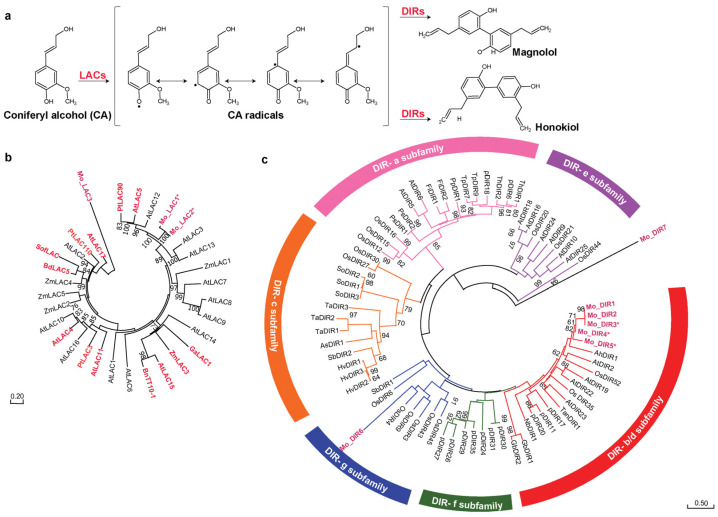
Proposed biosynthesis pathway for neolignans, honokiol and magnolol, in *Magnolia obovata.* (**a**) Proposed biosynthetic pathway. Abbreviations: LACs (laccases), and DIRs (dirigent proteins); Phylogenetic analysis of transcripts included in T12 module and annotated as LACs (**b**); and DIRs (**c**). The nucleotide sequences for candidate LACs and DIRs were translated and the corresponding protein sequences were aligned with LACs and DIRs spanning across different plant species using MUSCLE program. * DIRs annotated sequences with functional dirigent-like protein domain. The evolutionary history was inferred by using the Neighbor-Joining method with bootstrap values obtained after applying 1000 replications using the MEGA X program. Bootstrap values above 60% are shown here. The tree is drawn to scale, with branch lengths in the same units as those of the evolutionary distances used to infer the phylogenetic tree. The accession number of all the LACs and DIRs used for phylogenetic analysis are included as Appendix A.

**Table 1 ijms-26-01068-t001:** Assembly statistics for *Magnolia obovata* de novo transcriptome assembly based on three popular assemblers and their combination.

Assembler	Kmer	No. of Contigs	N50	Average Length	Median Length	Max Length
CLC	20	126,234	1126	719	408	15,296
Trinity	25	404,394	1230	734	403	15,776
SOAPdenovo	31	147,639	1239	720	384	15,241
	41	148,423	1257	715	373	15,334
	51	146,829	1212	687	355	15,635
	63	135,683	1171	661	334	15,335
	71	109,186	1214	695	353	15,321
	91	28,652	1415	936	650	12,222
CLC_Trinity_SOAPdenovo (kmer31) CD-HIT-EST	N.A.	284,165	1280	762	424	17,676

## Data Availability

The Illumina raw sequence reads, the de novo transcriptome assembly, annotations, and expression value for the unigenes have been deposited in NCBI’s Gene Expression Omnibus (GEO) and are available at GEO series accession number GSE275868.

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
