# Peer review of "Multi-Omics Analysis Decodes Biosynthesis of Specialized Metabolites Constituting the Therapeutic Terrains of *Magnolia obovata"

_ijms, 2025, doi:10.3390/ijms26031068_

Round 1

Reviewer 1 Report

Comments and Suggestions for Authors

There are some problems with the introduction that could be improved. When talking about the plant, I suggest rewriting it starting with the family Magnoliaceae, genus Magnolia and species Magnolia obovata. Biological activities should be reported only for the species M. obovata. In addition, give a brief presentation of the substances found in this species. The objectives of the work are clear and well defined. The methodology is described in detail.

Regarding the results, it is clear that the WCNA analysis was satisfactory, however, nothing was commented on for the following parts of the plant: central cylinder and floral bud. Why?

As this is a multidisciplinary work, unfortunately, I will not be able to collaborate with the transcriptome part. It becomes clear when reading the work the importance of using the integration of multiomic techniques. The work presents a series of results in both the metabolome and transcriptome parts, making it of great interest for publication.

It is necessary to review the bibliographic references in detail since the scientific names are not in italics.

Author Response

Reviewer 1

There are some problems with the introduction that could be improved. When talking about the plant, I suggest rewriting it starting with the family Magnoliaceae, genus Magnolia and species Magnolia obovata. Biological activities should be reported only for the species M. obovata. In addition, give a brief presentation of the substances found in this species.

Author’s response: Thank you for your insightful comments and suggestions. We have carefully revised the introduction in-line with the feedback provided in the reviewer’s document (ijms-3438073-review.pdf).

While we have adhered to the suggested structure, we opted to limit the introductory content on the family to maintain focus on the medicinal properties of Magnolia obovata. It is important to note that M. obovata and M. officinalis are both commonly used species within the Magnoliaceae family for the extraction of therapeutic compounds. As such, their medicinal properties overlap significantly, and many traditional medicine practices use either species interchangeably for similar therapeutic purposes. For this reason, we have referenced both M. obovata and M. officinalis as key species in our discussion of medicinal properties. We have also added one sentence specifically related to the metabolites reported for M. obovata, and have restructured few sentences for improved clarity.

All modifications in the introduction can be reviewed in the manuscript, where changes are tracked for your convenience.

The objectives of the work are clear and well defined. The methodology is described in detail.

Authors response: Thank you for your positive feedback regarding the clarity of the objectives and the detailed description of the methodology.

Regarding the results, it is clear that the WCNA analysis was satisfactory, however, nothing was commented on for the following parts of the plant: central cylinder and floral bud. Why?

Author’s response: Thank you for your thoughtful comment. In our study, no metabolite module specific to the central cylinder or floral bud were detected in the WCNA analysis. Additionally, the major tissue of interest, in terms of both metabolite presence and medicinal relevance, is the bark of Magnolia obovata. Given the focus of the study on the therapeutic properties associated with the bark, we chose to emphasize this tissue in the results section. While the central cylinder and floral bud were analyzed, they did not show significant or relevant findings, which is why they were not discussed further. We believe this approach ensures a more focused and meaningful presentation of the results and enhances the narrative of the manuscript.

As this is a multidisciplinary work, unfortunately, I will not be able to collaborate with the transcriptome part. It becomes clear when reading the work the importance of using the integration of multiomic techniques. The work presents a series of results in both the metabolome and transcriptome parts, making it of great interest for publication.

Author’s response: Thank you for your kind feedback and for acknowledging the importance of integrating multi-omics approaches in our work. The integration of these techniques is indeed a core aspect of our study, and we are glad it resonates with you as an essential part of the research.

It is necessary to review the bibliographic references in detail since the scientific names are not in italics.

Author’s response: Thank you for pointing out this issue with the bibliographic references. We have carefully reviewed and corrected the formatting of all scientific names in the references, ensuring that they are now italicized as per standard guidelines. We appreciate your attention to this detail and have made the necessary adjustments in the revised manuscript. We believe that the manuscript in this form would be acceptable to the reviewer.

Reviewer 2 Report

Comments and Suggestions for Authors

In this study, Rai et al. investigated, from an “omic” point of view, using a combination of untargeted metabolomic and transcriptomic approaches, the biosynthesis of pharmacologically active compounds produced by the deciduous tree Magnolia obovata

In their investigation, the authors identified several metabolites and candidate genes associated with neolignan biosynthesis, highlighting the molecular mechanisms behind the production of these metabolites. Through the use of recent high-performance technologies, including bioinformatic approaches, they showed different approaches for the exploration of secondary metabolism in medicinal plants.

This study is therefore very interesting in the context of chemical ecology, paving the way for the exploration of these important and unexplored ecosystems and opening new challenges in this field. However, in the following comments, I have minor suggestions to improve the text.

Minor Comments to the authors

Comment 1

The text itself is clearly written, and the English is good, even though there are some formatting errors (e.g., line 70, “de novo” should be in italics). 

Comment 2

References reported in the text have to be adjusted according to the journal style, usually reported at the end of the sentence (e.g., on Page 2, Line 50, references [4, 6] should be reported right before the dot). The same happens at line 90, where the references reported can be grouped at the end as [18-26] instead of “[18-20] and, to some extent, in the fruits [21], leaves [22] (add a comma after leaves) and flowers [23-26]”.). This is repeated along the text, so I suggest changing it accordingly. 

2

In this study, Rai et al. investigated, from an “omic” point of view, using a combination of untargeted metabolomic and transcriptomic approaches, the biosynthesis of pharmacologically active compounds produced by the deciduous tree Magnolia obovata. The introduction section is properly structured and very clear, starting with general information about the tree and describing the pharmacological properties of its different sections, widely in use in the Asian countries. A more chemically relevant section follows, where the type of pharmacologically bioactive components are listed and the main, still unanswered question that the authors want to address with this research is properly stated, that is to investigate the biosynthesis of these pharmacologically bioactive compounds and the mechanisms behind their synthesis. Answering this question is, in my opinion, an advancement in the field of chemical ecology and also with regards to the natural product pharmacology and pharmacognosy, fields that are strongly uncovering new discoveries and knowledge that attract and interest the scientific community. From a methodological point of view, the high-throughput technologies they use for this investigation are known and widely used; hence, even if my expertise is more in the field of metabolomics than transcriptomics, the workflow they have used to perform the experiments and to analyze the data is solid and already used by several other authors in the same field for similar investigations. Also, the bioinformatic approach they have used for the data analysis is based on widely used software and protocols; therefore, the scientific outcomes can be considered solid and reliable. My only concern was related to the identification of the 105 metabolites, but in the Table they provided in the supplementary material, there is all the information required to definitely assign the identities of these metabolites, even though a more traditional approach (not based on bioinformatic tools and comparison with online available databases) would require the use of external, synthetic standards to be run in order to verify the identities. However, in recent years, the use of the comparison with online available datasets and software is decreasing the use of this methodological approach, so I do not have particular comments or suggestions for this. Hence, my idea regarding the work is that it is very interesting in the context of chemical ecology, as well as in the fields of pharmacognosy and natural product chemistry, as it brings new knowledge in these fields. From an English point of view, the paper is clearly written and fluent; I only have minor suggestions to improve the text, mainly regarding the references and the formatting, to make it really precise and properly written. Minor Comments to the authors Comment 1 The text itself is clearly written, and the English is good, even though there are some formatting errors (e.g., “officinalis” (line 39) as well as “de novo” (line 70) should be in italics), to be checked throughout the entire text. Comment 2 References reported in the text have to be adjusted according to the journal style, usually reported at the end of the sentence (e.g., on Page 2, Line 50, references [4, 6] should be reported right before the dot). The same happens at line 90, where the references reported can be grouped at the end as [18-26] instead of “[18-20] and, to some extent, in the fruits [21], leaves [22] (add a comma after leaves) and flowers [23-26]”.). This is repeated along the text, so I suggest changing it accordingly. 

Author Response

Reviewer 2

In this study, Rai et al. investigated, from an “omic” point of view, using a combination of untargeted metabolomic and transcriptomic approaches, the biosynthesis of pharmacologically active compounds produced by the deciduous tree Magnolia obovata. The introduction section is properly structured and very clear, starting with general information about the tree and describing the pharmacological properties of its different sections, widely in use in the Asian countries. A more chemically relevant section follows, where the type of pharmacologically bioactive components are listed and the main, still unanswered question that the authors want to address with this research is properly stated, that is to investigate the biosynthesis of these pharmacologically bioactive compounds and the mechanisms behind their synthesis. Answering this question is, in my opinion, an advancement in the field of chemical ecology and also with regards to the natural product pharmacology and pharmacognosy, fields that are strongly uncovering new discoveries and knowledge that attract and interest the scientific community. From a methodological point of view, the high-throughput technologies they use for this investigation are known and widely used; hence, even if my expertise is more in the field of metabolomics than transcriptomics, the workflow they have used to perform the experiments and to analyze the data is solid and already used by several other authors in the same field for similar investigations. Also, the bioinformatic approach they have used for the data analysis is based on widely used software and protocols; therefore, the scientific outcomes can be considered solid and reliable. My only concern was related to the identification of the 105 metabolites, but in the Table they provided in the supplementary material, there is all the information required to definitely assign the identities of these metabolites, even though a more traditional approach (not based on bioinformatic tools and comparison with online available databases) would require the use of external, synthetic standards to be run in order to verify the identities. However, in recent years, the use of the comparison with online available datasets and software is decreasing the use of this methodological approach, so I do not have particular comments or suggestions for this. Hence, my idea regarding the work is that it is very interesting in the context of chemical ecology, as well as in the fields of pharmacognosy and natural product chemistry, as it brings new knowledge in these fields. From an English point of view, the paper is clearly written and fluent; I only have minor suggestions to improve the text, mainly regarding the references and the formatting, to make it precise and properly written.

Minor Comments to the authors Comment 1

The text itself is clearly written, and the English is good, even though there are some formatting errors (e.g., “officinalis” (line 39) as well as “de novo” (line 70) should be in italics), to be checked throughout the entire text.

Author’s comments: Thank you for your positive feedback on the clarity of the text. We appreciate your careful attention to the formatting details. We have reviewed the manuscript and corrected the instances where italics were missing, including officinalis (line 39) and de novo (line 70), as well as any other similar formatting issues throughout the entire text. These changes have been made in the revised manuscript.

Comment 2 References reported in the text have to be adjusted according to the journal style, usually reported at the end of the sentence (e.g., on Page 2, Line 50, references [4, 6] should be reported right before the dot).

The same happens at line 90, where the references reported can be grouped at the end as [18-26] instead of “[18-20] and, to some extent, in the fruits [21], leaves [22] (add a comma after leaves) and flowers [23-26]”.). This is repeated along the text, so I suggest changing it accordingly. 

Author’s comments: Thank you for your valuable feedback regarding the placement and grouping of references in the manuscript. We have carefully reviewed the text and adjusted the citation style where appropriate, ensuring that references are placed at the end of sentences as per the journal's guidelines.

However, in cases where references pertain to specific databases, such as KEGG, KNAPSACK, etc., we have retained the citation placement immediately following the mention of these databases to ensure clarity and accuracy. These references are essential for proper attribution and context, and we believe this placement enhances the readability and precision of the manuscript.

We appreciate your understanding of this approach and have made all other necessary adjustments to ensure consistent formatting throughout the manuscript.